# Influence of adiposity and physical activity on the cardiometabolic association pattern of lipoprotein subclasses to aerobic fitness in prepubertal children

Tarja Rajalahti[1,2,3], Eivind Aadland[4], Geir Kåre Resaland[4,5], Sigmund Alfred Anderssen[4,6], Olav Martin Kvalheim[1]*

1 Department of Chemistry, University of Bergen, Bergen, Norway, 2 Førde Health Trust, Førde, Norway, 3 Red Cross Haugland Rehabilitation Centre, Flekke, Norway, 4 Department of Sport, Food and Natural Sciences, Western Norway University of Applied Sciences, Sogndal, Norway, 5 Faculty of Education, Center for Physical Active Learning, Arts and Sports, Western Norway University of Applied Sciences, Sogndal, Norway, 6 Department of Sports Medicine, Norwegian School of Sport Sciences, Oslo, Norway

* olav.kvalheim@uib.no

**Data Availability Statement:** Data analyzed are provided as Excel files as Suppl. Inf. S1–S5 Tables.

## Abstract

Aerobic fitness (AF) and lipoprotein subclasses associate to each other and to cardiovascular health. Adiposity and physical activity (PA) influence the association pattern of AF to lipoproteins almost inversely making it difficult to assess their independent and joint influence on the association pattern. This study, including 841 children (50% boys) 10.2 ± 0.3 years old with BMI 18.0 ± 3.0 kg/m2 from rural Western Norway, aimed at examining the association pattern of AF to the lipoprotein subclasses and to estimate the independent and joint influence of PA and adiposity on this pattern. We used multivariate analysis to determine the association pattern of a profile of 26 lipoprotein features to AF with and without adjustment for three measures of adiposity and a high-resolution PA descriptor of 23 intensity intervals derived from accelerometry. For data not adjusted for adiposity or PA, we observed a cardioprotective lipoprotein pattern associating to AF. This pattern withstood adjustment for PA, but the strength of association to AF was reduced by 58%, while adjustment for adiposity weakened the association of AF to the lipoproteins by 85% and with strongest changes in the associations to a cardioprotective high-density lipoprotein subclass pattern. When adjusted for both adiposity and PA, the cardioprotective lipoprotein pattern still associated to AF, but the strength of association was reduced by 90%. Our results imply that the (negative) influence of adiposity on the cardioprotective association pattern of lipoproteins to AF is considerably stronger than the (positive) contribution of PA to this pattern. However, our analysis shows that PA contributes also indirectly through a strong inverse association to adiposity. The trial was registered 7 May, 2014 in clinicaltrials.gov with trial reg. no.: NCT02132494 and the URL is https://clinicaltrials.gov/ct2/results?term=NCT02132494&cntry=NO.

**Funding:** SAA, grant number 221047/F40, The Research Council of Norway, https://www.forskningsradet.no/, and GKR, grant number 1042294, The Gjensidige Foundation, https://www.gjensidigestiftelsen.no/, none of the funders played any role in the study design, data collection and analysis, decision to publish, or preparation of the manuscript.

**Competing interests:** The authors have declared that no competing interests exist.

## Introduction

Aerobic fitness (AF) is a strong predictor of cardiometabolic health in adults [1]. High concentration of small low-density lipoprotein (LDL) particles in adults correlates to cardiovascular disease (CVD) [2]. A pattern of high concentration of lipoprotein triglycerides (TG), very-low-density lipoproteins (VLDL) and large VLDL particles, and low concentration of high-density lipoproteins (HDL), large HDL and large LDL particles, and large average size of VLDL particles and low average size of HDL and LDL particle size correlates to insulin resistance [3–5]. This condition may ultimately translate into to type 2 diabetes mellitus and progress into CVD.

Associations between AF and cardiovascular risk factors are also observed for children [6] and unfavorable associations between adiposity, lipoproteins and cardiovascular risk factors have been verified in studies of children and adolescents. Thus, Suriano et al. [7] found waist circumference to be a predictor of cardiovascular risk in children, while Slyper et al. [8] found that the body mass index (BMI), and several lipoprotein features, correlated with thickening of the intima-media of the carotid artery in a cohort of adolescents. Cardiovascular status as expressed by AF and serum lipoprotein patterns in early years proceeds into adulthood with consequences later in life [9,10]. It is therefore important to study such associations for children and adolescents. In a previous study, we therefore investigated the association between AF and the lipoprotein pattern with BMI as covariate in 94 prepubertal children [11]. AF correlated positively to a cardioprotective lipoprotein pattern of high concentration of HDL, large and very large HDL particles and average size of HDL particles and to low concentrations of TG, chylomicrons (CM), VLDL, large and medium size VLDL particles, and average size of VLDL particles. BMI had a strong negative association to this pattern implying that control for adiposity is crucial for assessing the independent association pattern of lipoproteins to AF. In a recent investigation [12] of associations of adiposity, physical activity (PA) and lipoprotein subclasses for a cohort of 841 children, we found a similar pattern as for AF. The pattern withstood adjustment for adiposity.

Ekelund et al. [13] observed an independent association of PA and AF to metabolic risk factors in 1709 European children and PA/exercise intervention studies on adults [14,15] have shown strong impact of PA on the lipoprotein pattern. Considering those studies, our previous investigation on the association of AF to lipoprotein subclasses [11] was constrained by not including PA. In intervention studies for adults, increased PA correlated partly to the cardioprotective lipoprotein pattern to AF observed in our study for children, but PA in adults also correlated to reduced concentrations of LDL and the small atherogenic LDL particles and increased concentration of large LDL particles leading to increased average size of LDL particles. The same pattern was observed for physical active versus inactive adults by Kujala et al. [16] and the pattern withstood adjustment for adiposity. A meta- analysis including 10 intervention studies on adults [17] confirmed the strong favorable impact of PA on the lipoprotein subclass pattern.

In this work, our aim is to extract the association pattern of AF to lipoproteins in children and to examine the independent and joint influence of adiposity and PA on the association pattern. The same age group as in our previous study [11] is investigated, but for much larger cohort [18]. The relative influence of adiposity and PA on the association of AF to the lipoproteins is assessed through variance explained in AF and lipoproteins. Our aims present some challenges as adiposity and PA have virtually opposite association pattern to the lipoproteins [12], and the lipoprotein features are strongly multicollinear, as are the covariates adiposity and PA. The high-resolution PA descriptor used in this work even possessed linear dependency which requires a novel approach to adjustment to achieve the objectives of our study.

## Materials and methods

### Study and participants

1129 5th graders (94% of those invited) from 57 schools in Western Norway participated in this study [18]. Of these, 841 children provided valid baseline data on all relevant variables and were included in the present analysis.

Our procedures and methods conform to ethical guidelines defined by the World Medical Association's Declaration of Helsinki and its subsequent revisions. The South-East Regional Committee for Medical Research Ethics in Norway approved the study protocol (reference number 2013/1893). Prior to all testing, we obtained written informed consent from each child's parents or legal guardian and from the responsible school authorities. The study is registered in Clinicaltrials.gov with identification number: NCT02132494.

### Aerobic fitness test

The Andersen aerobic fitness test [19], which is a proxy for AF [20], but less influenced by adiposity than VO2$_{peak}$ in children [11], was executed according to the standard procedure. The test measured the total distance covered during a 10-minutes run. Children ran from one endline to another (20 m apart) in a to-and-fro movement intermittently, with 15-second work periods and 15-second breaks.

### Lipoprotein subclasses

Overnight fasting serum samples were obtained and stored at -80 ˚C according to a standardized protocol [21] and shipped on dry ice to the laboratories doing the analyses.

Serum lipoprotein profiles were characterized by 26 measures: Concentrations of total cholesterol (TC), total TG, CM, VLDL, LDL, HDL, two subclasses of CM (CM-1 and CM-2), five subclasses of VLDL (VLDL-L1, VLDL-L2, VLDL-L3, VLDL-M, VLDL-S), four subclasses of LDL (LDL-L, LDL-M, LDL-S, LDL-VS), six subclasses of HDL (HDL-VL1, HDL-VL2, HDL-L, HDL-M, HDL-S and HDL-VS), and the average particle size of VLDL, LDL and HDL. The subclasses are labelled according to the classification of Okazaki et al. [22] except that we have combined their three subclasses of very small LDL particles and their two subclasses of very small HDL particles. Following the terminology of Ozaki et al., the abbreviations VL, L, M, S and VS imply very large, large, medium, small, and very small particles. Some of the VL subclasses are divided further in accordance with the classification by Okazaki et al. We calculated triglyceride and cholesterol separately and independently for all subclasses using the approach described in the next paragraph but combined them into one subclass representing the total concentration for each subclass of lipoproteins.

The 26 lipoprotein measures were obtained from a partial-least-squares (PLS) regression model [23] obtained by calibrating proton nuclear magnetic resonance (NMR) spectra to results obtained from high performance liquid chromatography (HPLC). 106 serum samples were used in the calibration. Repeated Monte Carlo resampling was used to optimize the models with respect to predictive performance [24]. The HPLC analyses of the 106 calibration samples were performed by Skylight Biotech (Akita, Japan) as described by Okazaki et al. [22]. Proton NMR was performed at the MR core facility (NTNU, Trondheim) by a standard procedure [25] using a Bruker Avance III 600 MHz spectrometer, equipped with a QCI Cryo-Probe and an automated sample changer (SampleJet) (Bruker BioSpin GmbH, Karlsruhe, Germany.

## Physical activity descriptor

Raw PA data was obtained using the ActiGraph GT3X+ accelerometer [26] worn at the waist over seven consecutive days, except during water activities (swimming, showering) or while sleeping. Units were initialized at a sampling rate of 30 Hz. Files were analyzed at 1 second epochs using the KineSoft analytical software version 3.3.80 (KineSoft, Loughborough, UK). The use of 1 second epochs were found to be optimal for studying associations between metabolic and PA variables for this cohort [27]. Data were restricted to hours 06:00 to 23:59. In all analyses, consecutive periods of ≥ 60 minutes of zero counts were defined as non-wear time [28]. We applied wear time requirements of ≥ 8 hours/day and ≥ 4 days/week to constitute a valid measurement. We defined a PA descriptor by creating 23 PA variables of total time (min/day) obtained from the vertical axis to capture movement in narrow intensity intervals throughout the spectrum, from 0–99 to ≥ 10000 counts per minute (cpm). This descriptor covers the entire intensity spectrum. The intervals used for the descriptor were 0–99, 100–249, 250–499, 500–999, 1000–1499, 1500–1999, 2000–2499, 2500–2999, 3000–3499, 3500–3999, 4000–4499, 4500–4999, 5000–5499, 5500–5999, 6000–6499, 6500–6999, 7000–7499, 7500–7999, 8000–8499, 8500–8999, 9000–9499, 9500–9999 and ≥ 10000 cpm.

## Adiposity measures

We calculated three measures of adiposity: BMI ($kg/m^2$), waist to height ratio (WC/H), and skinfold thickness. BMI was calculated as mass divided by the squared height. Body mass was measured with an electronic scale (Seca 899, SECA GmbH, Hamburg, Germany). Height was measured with a transportable stadiometer (Seca 217, SECA GmbH, Hamburg, Germany). Waist circumference (WC) was measured twice between the lowest rib and the iliac crest with the child's abdomen relaxed at the end of a gentle expiration using an ergonomic measuring tape (Seca 201, SECA GmbH, Hamburg, Germany). If the difference between measurements was >1 cm, a third measurement was taken. The average of the two closest measurements was used for analyses. Skinfold thickness was measured at the left side of the body using a Harpenden skinfold caliper (Bull: British Indicators Ltd., West Sussex, UK). Two measurements were taken at each position (biceps, triceps, subscapular, and suprailiac). If the difference between measurements was >2 mm, a third measurement was obtained. The total sum of the average of the two closest measurements for each site was used for analysis.

## Transformations and pretreatment of variables

It is not a necessary assumption that the variables are normally distributed, but our method produces more stable models if the variables are approximately normally distributed. All variables, except age, the binary variable for sex, and the Andersen aerobic fitness test (which was approximately normally distributed), were thus log-transformed. Thereafter they were mean-centered and standardized to unit variance prior to the statistical analysis. The preprocessed data prior to centering and standardization to unit variance, are provided as supplementary information (S1 Table). After log transformation, normal probability plots showed that only CM, VLDL and a few of their subclasses in addition to TG still deviated from normal distribution.

## Data sets

Four different data set were created by using a projection approach [23] to adjust all variables for covariates.

Data set 1: Variables adjusted for age and sex (S2 Table).

Data set 2: Variables adjusted for age, sex, and PA (S3 Table).
Data set 3: Variables adjusted for age, sex, and adiposity (S4 Table).
Data set 4: Variables adjusted for age, sex, PA, and adiposity (S5 Table).

All variables, outcome, covariates, and explanatory variables were adjusted simultaneously and jointly by successive orthogonal projections [23]. Age and sex had only weak correlations to the covariates adiposity and PA and were adjusted for without requiring any special action, but for the linear dependent PA descriptor and the nearly dependent adiposity descriptor of BMI, W/H, and skinfold, we had to follow a different path. The PA descriptor of the 23 intensity variables was orthogonalized using principal component analysis (PCA) [23]. Monte Carlo resampling with 100 repetitions leaving out randomly and predicting 25% of the data in each run showed that four principal components, explaining jointly 92.8% of the total variance of the PA variables, contained all the predictive information about PA. We used these four orthogonal PCs to adjust for PA by successive orthogonal projections [23]. The same procedure applied for the three adiposity measures showed that only the first PC, accounting for 88.1% of the total variance in the adiposity variables, contained predictive information and was used to adjust for adiposity.

## Regression models

We used multivariate pattern analysis [29] with AF as outcome and the lipoproteins as explanatory variables for modelling of data set 1–4. The procedure uses PLS regression [23] with the number of PLS components determined by a significance test based on 1000 models calculated by repeated Monte Carlo resampling [24]. Post-processing of the PLS models with target projection (TP) [23] provides a single predictive vector for the lipoproteins quantifying the associations to the predicted AF. For model interpretation and visualization of association patterns, we used selectivity ratio (SR) plots [30].

## Results

### Descriptive statistics for the variables

The 841 children (50% boys) were (mean ± standard deviation) 10.2 ± 0.3 years old. The result for the Andersen test was 898 ± 102 m, for BMI 18.0 ± 3.0 kg/m2, for WC/H 0.43 ± 0.05, and for skinfold thickness 49.8 ± 26.4 mm. S6 Table provides mean and standard deviation for a range of common anthropometric and blood variables for this cohort. Mean and standard deviation for the 23 variables defining the PA descriptor is provided in S7 Table. S1 File in the present article contains mean and standard deviation for concentrations of lipoproteins and average particle size for VLDL, LDL and HDL.

### Correlations between variables

Correlations coefficients between the variables after adjustments for age and sex are tabulated in S8 Table and briefly summarized in S1 File. The correlation patterns of AF and the adiposity measures to the lipoprotein profile and PA are almost opposite, while AF and high-intensity PA shares a similar association pattern to lipoproteins.

### Models

Fig 1 shows the association patterns of AF to lipoproteins for the four models.

Heights of bars show quantitatively the associations of the lipoprotein features to the predicted AF. Bars above and below the vertical line at zero implies positive and negative associations to AF, respectively.

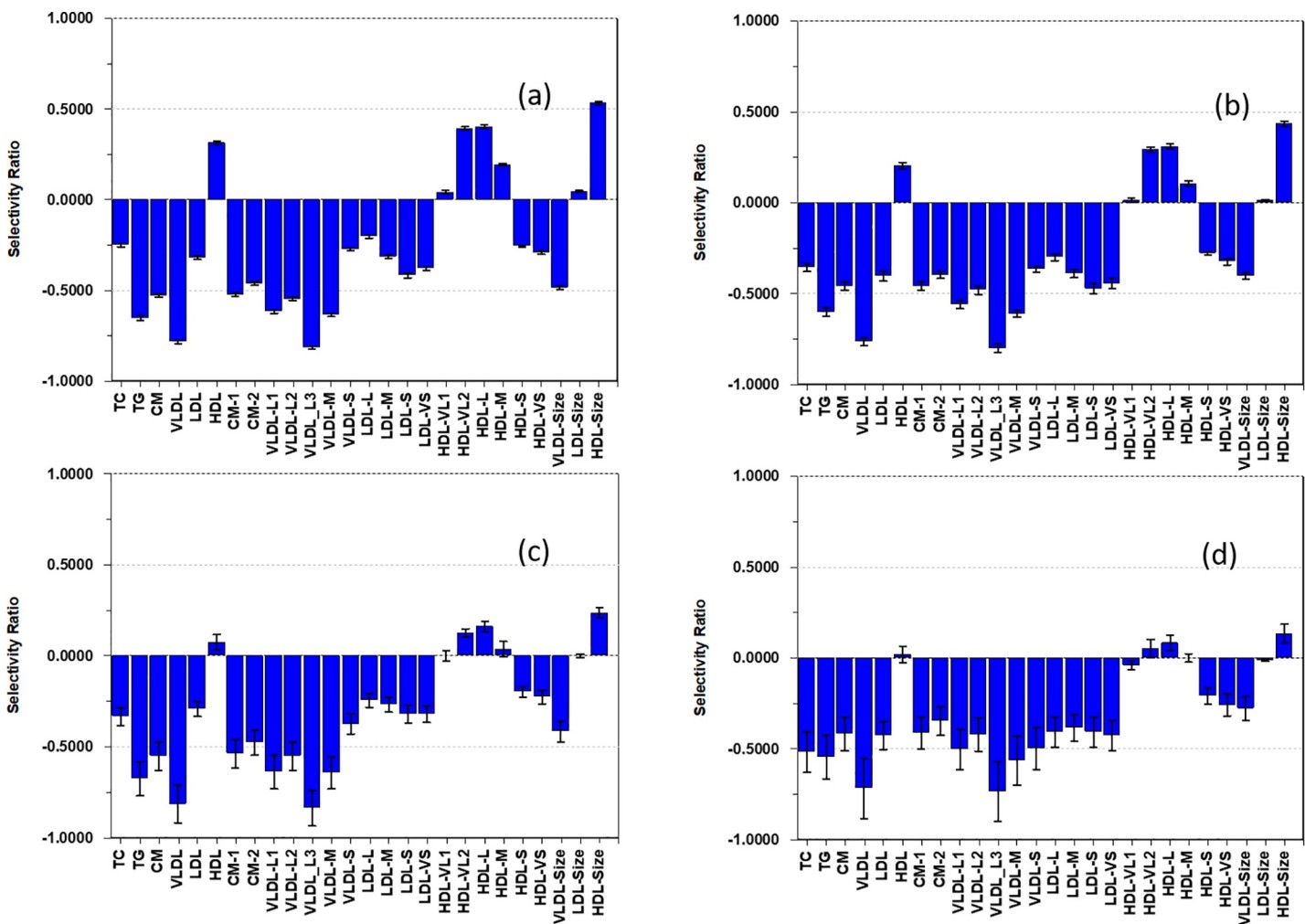

**Fig 1. Selectivity ratio plot of the models.** The Andersen aerobic fitness test is outcome and the lipoproteins features are explanatory variables. Adjustment for a) age and sex, b) age, sex, and PA, c) age, sex, and, adiposity, d) age, sex, PA, and adiposity. The error intervals on the bars correspond to 95% confidence limits.

Table 1 summarizes features of data and models for the association between AF and the lipoproteins.

## Discussion

Table 1 shows that adjustment has a larger influence on AF (column 1) than on the lipoproteins (column 2). For instance, adjustment for age and sex removes 8.9% of the variance in AF,

**Table 1. Description of data and models.**

| Data | R2AF₀[a] | R2LP₀[a] | R2AF[b] | R2LP[b] | SR plot |
|---|---|---|---|---|---|
| Adjusted for age, sex | 91.1 | 98.6 | 9.2 | 40.4 | Fig 1A |
| Adjusted for age, sex, PA | 73.0 | 95.0 | 3.9 | 37.5 | Fig 1B |
| Adjusted for age, sex, adiposity | 69.3 | 90.6 | 1.4 | 33.3 | Fig 1C |
| Adjusted for age, sex, adiposity, PA | 62.1 | 89.5 | 0.9 | 31.8 | Fig 1D |

[a]Percent remaining variance of total variance in aerobic fitness (AF) and lipoprotein (LP) profile after adjustment.

[b]Percent explained variance (by models) in the lipoproteins and AF of their total original variance.

but only 1.4% for lipoproteins. Boys were more physical active than girls in our cohort so adjustment for sex explains the reduction in variance in AF. The minor reduction of lipoprotein variance is expected as prepubertal boys and girls have similar lipoprotein profiles [31]. Adjustment for PA or adiposity reduces the variance in AF profoundly and of the same magnitude, but reduction in variance for lipoproteins is twice as large for adjustment of adiposity compared to PA implying a stronger confounding influence of adiposity than PA on the lipoprotein association pattern to AF. The reduction in explained variance in AF after adjustments (Table 1, column 3) confirms this expectation. Furthermore, adjustment for PA after adjustment for adiposity leads only to a minor additional reduction in explained variance for AF. This observation further implies a correlation between PA and adiposity so that adjustment for adiposity removes variance in PA and vice versa [12]. The strength of this association increases with increasing intensity of PA and peaks with a flat maximum around 7000–7500 counts/minute (S1 File). This pattern resembles the association pattern of PA to other metabolic health variables [32], but the maximum correlation is shifted towards higher PA intensity. Explained variance in lipoproteins (Table 1, column 4) relating to AF remains relatively robust to adjustments.

The SR plot of the model adjusted for age and sex (Fig 1A) reveals a positive association of AF to a cardioprotective lipoprotein pattern: Negative associations to concentrations of TG, CM, VLDL, and all their subclasses except VLDL-S, and to LDL and all LDL subclasses except LDL-L, and the small and very small HDL particles, and positive associations to HDL, HDL-VL2, and HDL-L. Association is also negative to average size of VLDL particles and positive to the average size of HDL particles. This pattern withstand adjustment for PA (Fig 1B), but adjustment for adiposity weakens the association of AF to the cardiometabolic favorable patterns of HDL particles. Further weakening is observed by adjustment for both adiposity and PA. However, all associations of AF to lipoprotein features found significant before adjustments, are still significant after adjustment as inferred from the 95% limits confidence bands in the SR plot.

The association of AF to the cardioprotective lipoprotein pattern resisted adjustment by both PA and adiposity, but the associations to total HDL and the very large and large HDL subclasses were further weakened (Fig 1D). Adiposity appears more important than PA for the association pattern of AF to the lipoproteins. This is consistent with previous investigations. Thus, in a cohort of 262 children (9–10 years old), Hager et al. [33] observed decrease in TG, TC and LDL and increase in HDL with increase in AF, but strong association of the lipoproteins to body fat led to the conclusion that "the goal of favorably altering blood lipids in children should begin with increasing PA and fitness, which in turn will lead to reductions in body fat". This recommendation was supported by Slyper et al. [8]. In a cohort of 61 obese non-diabetic adolescents, they observed a lipoprotein pattern comparable to ours associating to BMI and concluded that "focus of CVD prevention in the adolescent obese should be visceral obesity, and not blood lipids or lipid subclasses." Gutin et al. [34] examined relationships between AF, body fat, and lipoproteins, the latter being expressed as an atherogenic index, in a cohort of 57 children. They observed an inverse relationship between AF and CV risk but increased risk with body fat. In a cohort of 590 children, Hurtig-Wennlöf et al. [35] found positive association of AF to HDL and negative associations to TC and TG and stronger association of AF than PA to CV risk factors. Our finding that the association of AF to the cardiometabolic lipoprotein pattern resists adjustment for PA supports their observation. In a cohort of 1826 adolescents, Bell et al. [36] found similar associations to PA for a comprehensive lipoprotein profile as we observed to PA [12] for the same cohort as analyzed in the present work and for AF for the smaller cohort of 94 children [11] implying a similar association patterns for AF and PA. However, the strong negative association of LDL and all subclasses of

LDL to AF found in the present investigation was not observed for PA in Bell et al. [36] or in Rajalahti et al. [12]. Ekelund et al. [13] found independent effects from AF and PA on metabolic risk factors in children implying that AF associates to lipoproteins partly independently of PA. Several investigations on adults [14–17] have found similar association of PA to lipoproteins as we did for AF in the children. Hence, the observed association pattern with AF partly mirrors the correlation between PA and AF. Thus, we observed a decrease in the total LDL concentration and the small and very small LDL particles with increased AF in line with association pattern between PA and lipoproteins in adult populations [14–17]. However, also concentration of medium and large LDL particles decreases with increase in AF rendering average LDL particle size uncorrelated to AF in our study (Fig 1).

## Strengths and weaknesses

Studies of associations between AF and lipoproteins in children have mostly been limited to the standard lipid panel of TC, LDL and HDL cholesterol, and TG. Resolution into subclasses discriminates between small and large LDL and HDL particles, which associate inversely to cardiometabolic health, allowing an understanding of how AF impacts cardiovascular health through its association to lipoprotein pattern.

Cholesterol levels peak in prepubertal children at approximately 10 years age and then drop during puberty before rising again during adulthood [37]. For children, it is therefore beneficial to constrain such studies to a narrow age range.

Our analytical approach is adapted to handle multicollinear data and enables adjustment for confounders with linear dependency. This provides net association patterns and quantify the influence of confounders on the strength of the patterns which represent a challenge for use of molecular metabolomics descriptors in PA/exercise studies [38].

Because our analyses were restricted to cross-sectional associations, a limitation is that we cannot infer causality from our findings. Furthermore, our cohort embraces only Norwegian children. This limits the generalization of our study since there are differences in lipoprotein levels between different ethnic groups that may impact on the association to AF. However, in addition to studies discussed above, Okuma et al. [39] observed the same inverse association of adiposity to the cardioprotective subclass pattern of HDL in Japanese schoolchildren as observed in this study. So, the association patterns between lipoprotein and AF and its relation to PA and adiposity extend beyond the ethnic group in our study. Our study lacks information about diet which impacts the lipoprotein distribution. This is a limitation of the study design.

## Conclusion

Since cardiometabolic risk factors carry over from childhood to adulthood, it is crucial to understand the complex relationships between AF, lipoproteins, adiposity, and PA in children. Our study shows that AF associates positively to a cardioprotective lipoprotein pattern but that the strength of this association is strongly influenced by PA and adiposity. PA associates positively to both AF and this pattern, while adiposity associates almost inversely and stronger than PA to this pattern. However, since PA and adiposity are inversely associated, adjustment by adiposity also removes variance of PA shared with AF. Thus, adiposity and PA in childhood influences the cardioprotective lipoprotein pattern directly, but also indirectly through the inverse relation of PA to adiposity. Although adiposity has a stronger independent association than PA to cardiometabolic health, the indirect influence of PA through the inverse relationship to adiposity must be taken into consideration when assessing the relative importance of these two factors on cardiometabolic health. Physical activity has an additional positive effect by preventing increase in adiposity and thus strengthening the cardiometabolic healthy lipoprotein pattern.

## Supporting information

**S1 Table. Unadjusted data.** Data for 841 children where 7 children have replicated lipoprotein analyses (implied by sample label containing R). Note that to prevent possibilities for identification, the variable AGE has been rounded to integers in the table, while it was used with two decimal places in the actual calculations.
(XLSX)

**S2 Table. Data adjusted for age and sex.** Data for 841 children where 7 have replicated lipoprotein analysis (implied by sample label containing R).
(XLSX)

**S3 Table. Data adjusted for age, sex, and physical activity.** Data for 841 children where 7 have replicated lipoprotein analysis (implied by sample label containing R).
(XLSX)

**S4 Table. Data adjusted for age, sex and adiposity.** Data for 841 children where 7 have replicated lipoprotein analysis (implied by sample label containing R).
(XLSX)

**S5 Table. Data adjusted for age, sex, physical activity, and adiposity.** Data for 841 children where 7 have replicated lipoprotein analysis (implied by sample label containing R).
(XLSX)

**S6 Table. Mean and standard deviation for demography, anthropometry and metabolic health for the children.**
(DOCX)

**S7 Table. Time spent in different PA intensity intervals (mean (SD)) with epoch setting 1s.**
(DOCX)

**S8 Table. Correlation coefficients, Correlations between the Andersen test, adiposity, the lipoproteins and the PA variables after adjustment for age and sex.**
(XLSX)

**S1 File. Summary statistics for lipoproteins.** Mean and standard deviation of lipoprotein features and their correlations with aerobic fitness (AF), physical activity (PA) and adiposity measures.
(DOCX)

## Acknowledgments

We thank all children, parents, and teachers at the participating schools for their excellent cooperation during the data collection, and Trygve Andreassen and Tone F. Bathen at the MR Core Facility, Department of Circulation and Medical Imaging, NTNU—Norwegian University of Science and Technology, Trondheim, Norway for help with the NMR analysis. Turid Skrede, Mette Stavnsbo, Katrine Nyvoll Aadland, Øystein Lerum, Einar Ylvisåker, and students at the Western Norway University of Applied Sciences, are thanked for assisting in data collection.

## Author Contributions

**Conceptualization:** Tarja Rajalahti, Eivind Aadland, Geir Kåre Resaland, Sigmund Alfred Anderssen, Olav Martin Kvalheim.

**Data curation:** Tarja Rajalahti, Eivind Aadland.

**Formal analysis:** Olav Martin Kvalheim.

**Funding acquisition:** Geir Kåre Resaland, Sigmund Alfred Anderssen.

**Investigation:** Tarja Rajalahti, Eivind Aadland, Geir Kåre Resaland.

**Methodology:** Olav Martin Kvalheim.

**Validation:** Olav Martin Kvalheim.

**Visualization:** Tarja Rajalahti.

**Writing – original draft:** Olav Martin Kvalheim.

**Writing – review & editing:** Tarja Rajalahti, Eivind Aadland, Geir Kåre Resaland, Sigmund Alfred Anderssen.

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
