## [Decision Letter · Decision Letter 0]

17 Sep 2021

PONE-D-21-18590

Influence of adiposity and physical activity on the cardiometabolic association pattern of lipoprotein subclasses to aerobic fitness in prepubertal children

PLOS ONE

Dear Dr. Kvalheim,

Thank you for submitting your manuscript to PLOS ONE. After careful consideration, we feel that it has merit but does not fully meet PLOS ONE’s publication criteria as it currently stands. Therefore, we invite you to submit a revised version of the manuscript that addresses the points raised during the review process.

We look forward to receiving your revised manuscript.

Kind regards,

Martin Senechal, PhD

Academic Editor

PLOS ONE

“The study was funded by the Research Council of Norway (grant number 221047/F40) and the Gjensidige Foundation (grant number 1042294). None of the funding agencies had any role in the study design, data collection, analysis or interpretation of the data, or in writing the manuscripts.”

“SAA, grant number 221047/F40, The Research Council of Norway, https://www.forskningsradet.no/, and GKR, grant number 1042294, The Gjensidige Foundation, https://www.gjensidigestiftelsen.no/, none of the funders played any role in the study design, data collection and analysis, decision to publish, or preparation of the manuscript.”

Additional Editor Comments (if provided):

Reviewers' comments:

Reviewer's Responses to Questions

**Comments to the Author**

1. Is the manuscript technically sound, and do the data support the conclusions?

Reviewer #1: Partly

Reviewer #2: Partly

2. Has the statistical analysis been performed appropriately and rigorously? 

Reviewer #1: I Don't Know

Reviewer #2: Yes

3. Have the authors made all data underlying the findings in their manuscript fully available?

Reviewer #1: Yes

Reviewer #2: Yes

4. Is the manuscript presented in an intelligible fashion and written in standard English?

Reviewer #1: Yes

Reviewer #2: No

5. Review Comments to the Author

Reviewer #1: Important note: This review pertains only to ‘statistical aspects’ of the study and so ‘clinical aspects’ [like medical importance, relevance of the study, ‘clinical significance and implication(s)’ of the whole study, etc.] are to be evaluated [should be assessed] separately/independently. Further please note that any ‘statistical review’ is generally done under the assumption that (such) study specific methodological [as well as execution] issues are perfectly taken care of by the investigator(s). This review is not an exception to that and so does not cover clinical aspects {however, seldom comments are made only if those issues are intimately / scientifically related & intermingle with ‘statistical aspects’ of the study}. Agreed that ‘statistical methods’ are used as just tools here, however, they are vital part of methodology [and so should be given due importance].

COMMENTS: Your ABSTRACT is well drafted but assay type. Please note that it is preferable to divide the ABSTRACT with small sections like ‘Objective(s)’, ‘Methods’, ‘Results’, ‘Conclusions’, etc. which is an accepted practice of most of the good/standard journals [including this one]. It will definitely be more informative then, I guess, whatever the article type may be.

“Why [refer to ‘Availability of data’ section on page 14] the data analyzed in this article are added as Suppl. Mat.”? Moreover, note that most of the ‘Correlations of AF, lipoproteins, physical activity and adiposity’ as reported/displayed in Suppl. Mat. 2 are generally ‘low’. In this context, please read the following [which is pasted from one standard textbook on ‘Research Methodology’]. I am sure that the authors already know these things, however, it is very essential to keep the limitations in mind while interpreting results {which unfortunately are not interpretated adequately}.

Statistical test usually used to assess significance of Pearson’s ‘Correlation coefficient (r)’ is ‘t’ [where t = { r � [(n-2) / (1-r2)] }for df=n-2, n is sample size] and here Ho is that the population/standard value of ‘r’ is zero. You need r=0.878 to be significant at 5% when n=5 but you need r=0.273 only, if n=50 & you need r=0.088 only, if n=500. Because ‘P-value’ heavily depends on sample size, it is customary to use the (available in most text books on ‘Biostatistics’ or on ‘www/net’) guidelines [very strongly suggesting] to consider an absolute value of ‘Correlation coefficient’ for interpreting positive or negative correlations (and do not rely only on corresponding ‘P’-value but also consider an absolute value of ‘Correlation coefficient’). [This argument is equally applicable to non-parametric Spearman’s ‘Correlation coefficient (ρ)’ as well.]

I request the authors to confirm the level of measurement of “Aerobic fitness test” scores [as said in concerned section that the test measured the total distance covered during a 10-minutes run. {and that Children ran from one end-line to another (20 m apart) in a to-and-fro movement intermittently, with 15-second work periods and 15-second breaks}. Moreover, you stated that “We used a PA descriptor of 23 intervals defined by counts per minute (cpm) covering the entire intensity spectrum”. In this context (just reminder), please read the following [which is again pasted from the same textbook on ‘Research Methodology’].

Though the measures/tools used are appropriate, most of them yield data that are in [at the most] ‘ordinal’ level of measurement [and not in ratio level of measurement for sure {as the score two times higher does not indicate presence of that parameter/phenomenon as double (for example, a Visual Analogue Scales VAS score or say ‘depression’ score)}]. Then application of suitable non-parametric test(s) is/are indicated/advisable [even if distribution may be ‘Gaussian’ (i.e. normal)]. Agreed that there is/are no non-parametric test(s)/technique(s) available to be used as alternative in all situation(s) [suitable / most desired/applicable], but should be used whenever/wherever they are available.

It is true that variable ‘time’ is ‘ratio’ but the procedure to record it could change level of measurement. One should just watch-out though it is definitely appreciable that “All variables were thus log-transformed, mean-centered and standardized to unit variance prior to the statistical analysis. After log transformation, normal probability plots showed that only CM, VLDL and a few of their subclasses in addition to TG still deviated from normal distribution”.

Further I request the authors to read the article {just for information, authors are likely to have done this already} titled ‘Relative Importance of Borderline and Elevated Levels of Coronary Heart Disease Risk Factors’ (Ann Intern Med. 2005;142:393-402) & kindly note that I am not asking you to change the study design, but frankly speaking, ‘clinical aspects’ [like medical importance, relevance of the study, ‘clinical significance and implication(s) of the whole study, etc.] are not very clear and so in my considered opinion, should be assessed separately/independently.

Reviewer #2: The subject is of interest as these molecules may aid in pinpointing the molecular pathways linking health and disease in children and how they are influenced by lifestyle behaviours, such as physical activity, fitness and levels of adiposity. However, there are several issues described below that limit the interpretation of the findings generated

* I would consider renaming the title: Independent associations of aerobic fitness, physical activity and adiposity with lipoprotein subclasses in pre-pubertal children and also report the separate associations for PA and adiposity to the lipoprotein subclasses. This could be included as supplementary material. In my opinion, this is a more thorough approach with greater transparency.

* In the abstract, please include some basic information regarding your population. For example, the number of participants included, average age, BMI, waist circumference etc. It would also be helpful for the reader for this to be included in a Table in the main manuscript.

* The opening paragraph of the introduction contains one very long sentence. Consider revising.

* Please make it clear in the second paragraph of the introduction that you are referring to your previous work. The way it is worded presently implies that you are referring to the current study

* The last two sentences of the introduction would be better placed in the discussion

* More information is required regarding the accelerometers. For example, how did you define non-wear time? How many valid days were required as a minimum? It is also important to adjust for wear-time in the analysis. You also state that time spent in PA intensities obtained from the vertical axis was calculated for the subjects. Please include this information

* As a minimum, you need to state that your analysis is observational, meaning that you cannot prove biological mechanisms or demonstrate causality; reverse causality is also a possibility whereby those with a greater burden of obesity may be less likely to engage in greater volumes or intensities of physical activity

* Assuming this data is not available, the lack of adjustment for important confounders (i.e. diet quality, birthweight and socioeconomic status) needs to be addressed.

* The resolution for Figure 1 is too low to review

6. PLOS authors have the option to publish the peer review history of their article (what does this mean?). If published, this will include your full peer review and any attached files.

Reviewer #1: No

Reviewer #2: No

---

## [Author Response · Author response to Decision Letter 0]

3 Oct 2021

See letter with response to reviewers.

---

## [Decision Letter · Decision Letter 1]

19 Oct 2021

PONE-D-21-18590R1

Influence of adiposity and physical activity on the cardiometabolic association pattern of lipoprotein subclasses to aerobic fitness in prepubertal children

PLOS ONE

Dear Dr. Kvalheim,

Thank you for submitting your manuscript to PLOS ONE and thank you for addressing most of the comments from previous reviewers. We believe you should consider one comment raised by reviewer two, who feels that it could be better addressed. Also, as part of PLOS ONE publication criteria, make sure the manuscript meets the requirements from reporting guidelines. We feel that this study is reported in a way that does not fully meet those requirements. Therefore, we invite you to submit a revised version of the manuscript that addresses the points raised during the review process.

We look forward to receiving your revised manuscript.

Kind regards,

Martin Senechal, PhD

Academic Editor

PLOS ONE

Journal Requirements:

Reviewers' comments:

Reviewer's Responses to Questions

**Comments to the Author**

1. If the authors have adequately addressed your comments raised in a previous round of review and you feel that this manuscript is now acceptable for publication, you may indicate that here to bypass the “Comments to the Author” section, enter your conflict of interest statement in the “Confidential to Editor” section, and submit your "Accept" recommendation.

Reviewer #1: (No Response)

Reviewer #2: (No Response)

2. Is the manuscript technically sound, and do the data support the conclusions?

Reviewer #1: (No Response)

Reviewer #2: Yes

3. Has the statistical analysis been performed appropriately and rigorously? 

Reviewer #1: (No Response)

Reviewer #2: Yes

4. Have the authors made all data underlying the findings in their manuscript fully available?

Reviewer #1: (No Response)

Reviewer #2: Yes

5. Is the manuscript presented in an intelligible fashion and written in standard English?

Reviewer #1: (No Response)

Reviewer #2: Yes

6. Review Comments to the Author

Reviewer #1: COMMENTS: Most of the comments made on earlier draft by me (and hopefully by other respected reviewers also) were/are attended [though I am not convinced about few arguments & the way they are made]. I recommend the acceptance as the manuscript now (even earlier I accepted/appreciated the potential of this article) has achieved acceptable level, in my opinion.

Reviewer #2: The authors have done a good job of addressing previous comments. However, I still think the tables including PA descriptors and cardiometaboilc info should be included in this manuscript. It seems a little unrealistic to expect the reader to look elsewhere for this rudimentary information.

7. PLOS authors have the option to publish the peer review history of their article (what does this mean?). If published, this will include your full peer review and any attached files.

Reviewer #1: No

Reviewer #2: No

---

## [Author Response · Author response to Decision Letter 1]

21 Oct 2021

We have included the table requested by reviewer 2 and the editor.

---

## [Editor Report · Decision Letter 2]

29 Oct 2021

Influence of adiposity and physical activity on the cardiometabolic association pattern of lipoprotein subclasses to aerobic fitness in prepubertal children

PONE-D-21-18590R2

Dear Dr. Kvalheim,

We’re pleased to inform you that your manuscript has been judged scientifically suitable for publication and will be formally accepted for publication once it meets all outstanding technical requirements.

Kind regards,

Martin Senechal, PhD

Academic Editor

PLOS ONE

---

## [Editor Report · Acceptance letter]

9 Nov 2021

PONE-D-21-18590R2 

Influence of adiposity and physical activity on the cardiometabolic association pattern of lipoprotein subclasses to aerobic fitness in prepubertal children 

Dear Dr. Kvalheim:

I'm pleased to inform you that your manuscript has been deemed suitable for publication in PLOS ONE. Congratulations! Your manuscript is now with our production department. 

Kind regards, 

on behalf of

Dr. Martin Senechal 

Academic Editor

PLOS ONE